# Geo-Climatic Factors of Malaria Morbidity in the Democratic Republic of Congo from 2001 to 2019

**DOI:** 10.3390/ijerph19073811

**Published:** 2022-03-23

**Authors:** Eric Kalunda Panzi, Léon Ngongo Okenge, Eugénie Hamuli Kabali, Félicien Tshimungu, Angèle Keti Dilu, Felix Mulangu, Ngianga-Bakwin Kandala

**Affiliations:** 1Département de la Santé Communautaire, Institut Supérieur des Techniques Médicales de Kinshasa, B.P. 774, Kinshasa XI, Mont Ngafula, Kinshasa, Democratic Republic of the Congo; eric.panzi53@gmail.com (E.K.P.); okengeleon@yahoo.fr (L.N.O.); eugekab@gmail.com (E.H.K.); ftkatshidikaya@gmail.com (F.T.); 2Ministère de la Santé, Secrétariat Général /Cellule Suivi et Evaluation, 36 Avenue de la Justice Gombe, B.P. 3088, Kinshasa, Democratic Republic of the Congo; angeledilu@gmail.com; 3Ministère de la Santé Publique, Direction de la Surveillance Epidémiologique, 36 Avenue de la Justice Gombe, Kinshasa, Democratic Republic of the Congo; felixmulangu@gmail.com; 4Division of Epidemiology and Biostatistics, School of Public Health, University of the Witwatersrand, Johannesburg 2193, South Africa; 5Warwick Medical School, University of Warwick, Coventry CV4 7AL, UK

**Keywords:** poisson generalized linear model, malaria cases, morbidity, geo-climatic factors, DRC

## Abstract

Background: Environmentally related morbidity and mortality still remain high worldwide, although they have decreased significantly in recent decades. This study aims to forecast malaria epidemics taking into account climatic and spatio-temporal variations and therefore identify geo-climatic factors predictive of malaria prevalence from 2001 to 2019 in the Democratic Republic of Congo. Methods: This is a retrospective longitudinal ecological study. The database of the Directorate of Epidemiological Surveillance including all malaria cases registered in the surveillance system based on positive blood test results, either by microscopy or by a rapid diagnostic test for malaria was used to estimate malaria morbidity and mortality by province of the DRC from 2001 to 2019. The impact of climatic factors on malaria morbidity was modeled using the Generalized Poisson Regression, a predictive model with the dependent variable Y the count of the number of occurrences of malaria cases during a period of time adjusting for risk factors. Results: Our results show that the average prevalence rate of malaria in the last 19 years is 13,246 (1,178,383–1,417,483) cases per 100,000 people at risk. This prevalence increases significantly during the whole study period (*p* < 0.0001). The year 2002 was the most morbid with 2,913,799 (120,9451–3,830,456) cases per 100,000 persons at risk. Adjusting for other factors, a one-day in rainfall resulted in a 7% statistically significant increase in malaria cases (*p* < 0.0001). Malaria morbidity was also significantly associated with geographic location (western, central and northeastern region of the country), total evaporation under shelter, maximum daily temperature at a two-meter altitude and malaria morbidity (*p* < 0.0001). Conclusions: In this study, we have established the association between malaria morbidity and geo-climatic predictors such as geographical location, total evaporation under shelter and maximum daily temperature at a two-meter altitude. We show that the average number of malaria cases increased positively as a function of the average number of rainy days, the total quantity of rainfall and the average daily temperature. These findings are important building blocks to help the government of DRC to set up a warning system integrating the monitoring of rainfall and temperature trends and the early detection of anomalies in weather patterns in order to forecast potential large malaria morbidity events.

## 1. Introduction

Malaria is caused by infection with *protozoan* parasites of the species *Plasmodium*. *Plasmodium falciparum* is widespread in Africa, while *P. vivax*, *P. ovale*, and *P. malariae* infections are less common and geographically limited [1,2]. Malaria is a vector-borne disease whose existence and transmission depend on three main factors: The Plasmodium parasite, the Anopheles vector and the human host. Beyond these essential factors, the risk of malaria transmission can be maintained or reinforced by environmental or climatic conditions as well as socio-economic factors.

In the past, several authors have highlighted the existing relationship between climatic variations (rainfall, temperature and relative humidity) and malaria endemicity [3].

The difficulties in identifying, quantifying and predicting the effects of climate change on health relate to the characteristics of the “exposure” and to the development of causal relationships that are often complex and indirect and occur at different spatial and temporal scales. Several types of recommendations are proposed such as site sanitation and chemoprevention of seasonal malaria [4].

Establishing a relationship between the environment and health still remains a challenge for scientific knowledge because the causal chain between environmental factors and the resulting disease is complex. To test these causal hypotheses, it will be necessary to scrutinize causal mechanisms at multiple scales including social, ecological and gene-environment interactions [5].

Globally, the number of malaria cases is estimated to be 228 million in 2018 (95% confidence interval [CI]: 206–258 million), compared with 251 million in 2010 (95% CI: 231–278 million) and 231 million in 2017 (95% CI: 211–259 million) [6].

Each year, more than 4 million children worldwide die from environmentally-related diseases [7,8]. Malaria, diarrhea, and acute respiratory infections are the most significant environmental health problems, contributing to 26% of child deaths worldwide. These health problems are sometimes compounded by malnutrition and measles, which dangerously weaken children [9].

Climatic factors are an important determinant of various vector-borne diseases, many enteric diseases, and some waterborne diseases. This study aims to identify geo-climatic factors predictive of malaria prevalence from 2001 to 2019 in the Democratic Republic of Congo. Previous studies examining the relationship between climate and malaria have found lagged associations between climate variables (temperature and rainfall) and malaria cases over periods ranging from weeks to months [10,11].

Using mathematical modeling we link climatic factors and malaria prevalence adjusting for spatio-temporal variations to aid policymakers in making more responsible decisions. The results of our study may aid the DRC’s government to implement a weather-health alert system to effectively combat malaria and other diseases with epidemic potential, which is a major challenge to be met in DRC to make the existing epidemiological surveillance highly informative and intelligent.

## 2. Materials and Methods

### 2.1. Study Areas and Population

This study was conducted in the Democratic Republic of Congo (DRC) at the subnational level of 26 administrative provinces (See Figure 1). The DRC is located in the heart of Africa and is one of the largest countries on the continent with an area of 2,345,000 km^2^. It shares 9165 km of borders with nine neighboring countries: The Republic of Congo to the west, Uganda, Burundi, Rwanda and Tanzania to the east, the Central African Republic and Sudan (to the north), and Zambia and Angola to the south. The country’s extensive borders, combined with a lack of transport and communication infrastructure, make it particularly difficult to trade and move goods and people within the country.

The United Nations World Population Prospects estimates the 2021 population of DRC to 93,348,823 inhabitants with a population growth rate of 3.19%, the highest in the world and the population projected to surpass 100 million in 2024 and double its population by 2047. The fertility rate in DRC of 6.11 births per woman is also among the highest in the world. However, since the invasion of DRC in 1996 by neighboring countries mainly Rwanda and Uganda and the second war of invasion in 1998, also called the African world war, the deadliest conflict on earth since the Chinese Civil War with an estimated over 6 million civilian deaths from 1993 to 2003 alone and 47% of the deaths were children under five. In the economic war in DRC, which is still ongoing, 90% of soldiers died from malaria and disease, has displaced a huge percentage of the population, which makes the DRC one of the most challenging environments for health development in Africa with many issues that a rapidly increasing population can exacerbate such as ongoing conflict, lack of modern health services in addition to major health problems, the EBOLA pandemic, HIV/AIDs as well as rape, human trafficking and child labor [12,13].

### 2.2. Datasets

This is a retrospective longitudinal ecological study. We exploited the Mettelsat database for meteorological data on the one hand and the Directorate of Epidemiological Surveillance (DES) database including malaria data from 2001 to 2019 on the other, which has national coverage. The Directorate of Epidemiological Surveillance (DES) is centralizing the data on malaria of 26 Provincial Health Directorates (PHD) of the DRC and we used data from 2001 to 2019. Climatic data, including total rainfall quantity, temperature, wind speed, total evaporation under shelter, and average relative humidity, were measured on a permanent basis using different weather stations in the DRC. The distribution of these stations takes into account the variability of the different eco-zones with respect to their climate, their proximity to rivers and altitude. Geographic data are collected at the National Geographic Institute. Longitude, latitude, altitude, seasons and hydrography were also collected.

The sampling is three-stage random. To constitute our sample, we proceeded in the following way: In the first stage, we drew lots from the provincial health directorates with at least one weather station. In total, the DRC has 26 Provincial Health Directorates (PHD) and 519 Health Zones. For the malaria data, 13,332,072 observations were selected. These are 26 provinces × 19 years’ × 52 epidemiological weeks’ × 519 Health Zones. In the second stage, we selected the health zones with meteorological data. We reduced the observations by grouping them into 26 provinces, 19 years, 52 epidemiological weeks. Then the outliers and incomplete data were removed. In the third stage, we have the malaria cases completely and correctly recorded in the database. Then the data were aggregated into 494 observations (26 provinces multiplied by 19 years). This allowed us to forecast malaria prevalence to 2036.

### 2.3. Climatic Parameters

Climate is the average of weather conditions (temperature, precipitation, wind, humidity, air pressure) at a given location over a long period of time, usually 30 years. The relative humidity is the quantity of water vapor contained in a given volume of air compared to the maximum it could contain at a given temperature and pressure [12]. It is expressed as a percentage.

Precipitation is all meteoric water that falls on the surface of the earth, both in liquid form (drizzle, rain, downpour) and in solid form (snow, sleet, hail) and precipitation deposited or hidden (dew, white frost, frost). They are caused by a change in temperature or pressure. Precipitation is the only “input” to the main continental hydrological systems that are the watersheds. It is expressed in millimeters (mm) [12].

Temperature is an essential parameter that conditions all physiological activities and chemical reactions (role of temperature in the reactivation of cambium and the lifting of dormancy, in the inhibition of photosynthesis…). The air temperature depends on the solar radiation, the pressure of the atmosphere and its gas composition. Variations in air temperature are strongly buffered by atmospheric humidity, and it is in arid zones that the greatest daily thermal amplitudes are observed. By convention, the thermometer is placed inside an enclosure made of insulated material, painted white, placed at 1.5 m from the ground, the goal being to limit the solar radiation reflected by the ground. The temperature is expressed in degrees Celsius (°C) or degrees Fahrenheit (°F) [12].

Wind is the consequence of air movement. It is the result of pressure forces. It is characterized by its speed, generally expressed in kilometers per hour (km/h), or in knots (kts) in meteorological circles, and its direction or origin, indicated in degrees with respect to the North or with the help of a wind rose. The number of rainy days: In addition to the annual and seasonal totals, the frequency of rainy events is also an important parameter to take into account for the analysis of climatic conditions for the growth of forest stands (12).

Evaporation is the process by which an element passes from the liquid state to the gaseous state. In the context of the water cycle, this process implies the complementary process of condensation of water vapor contained in the atmosphere and its return to earth in the form of precipitation [12].

### 2.4. Statistical Methods

Let $Y_{i}$ be the number of malaria cases recorded in the i-th year, for i=1,…,n=19,where i=1  represents the year 2001 and i=19 represents the year 2019. Assume that $Y_{i}$ is generated according to a generalized Poisson model [14]. i.e.,
(1)$$\;Y_{i}I\alpha ,\mu_{i}∽\mathrm{GPoisson}\left(\alpha ,\mu_{i}\right)\;$$
where $\mu_{i}\;$ is the expected value and $\mu_{i}\;\mathrm{is}\;a\;\mathrm{dispersion}\;\mathrm{parameter}\;,\;\mathrm{for}\;i=1,\;\ldots ,n.$ Its probability function is given by:(2)$$P(Y_{i}=y_{i}I\alpha ,\mu_{i})=\left(\frac{\mu_{i}}{1+\alpha \mu_{i}}\right)\;\frac{{\left(1+\alpha ,y_{i}\right)}^{y_{i}-1}}{y_{i}\;!}exp\left\{\frac{-\mu_{i}\left(1+\alpha ,y_{i}\right)}{1+\alpha \mu_{i}}\right\}\;$$

For i=1,….,n.

Model (1) is a generalization of the Poisson distribution that allows the modeling of overdispersed or underdispersed data. When the dispersion parameter α=0, the probability function reduces to the probability function of the Poisson distribution. When α>0, the GPoisson distribution can be used for modeling data with overdispersion, and when α < 0, this distribution can be used for modeling data with underdispersion.

In addition, consider the interest in verifying whether there is a link between the number of malaria cases Y and the following climatic variables:$X_{1}$ = Total evaporation under shelter;$X_{2}$ = Mean relative humidity;$X_{3}$ = Maximum temperature;$X_{4}$ = Minimum temperature;$X_{5}$ = Mean wind speed at 2 m above ground$X_{6}$ = Season

Thus, consider y = $(y_{1}\ldots ,\;y_{n}{)}^{\prime }\;$ be the observed vector, of dimension $n\times 1\;;x\;\mathrm{be}\;\mathrm{the}\;\mathrm{matrix}\;\mathrm{of}\;\mathrm{observed}\;\mathrm{values}\;\mathrm{for}\;\mathrm{environmental}\;\mathrm{variables},\;\mathrm{of}\;\mathrm{dimension}\;n\times 7,\;\mathrm{and}\;x_{i}$ = $(1,\;x_{i1},$ $x_{i2},\;x_{i3},$ $x_{i4},$ $x_{i5},\;x_{i6}{)}^{\prime }$ be a row vector of *x*, for $y_{1}\in \left\{0,1,2,\;.\;.\;.\;\right\}\;$ and i=1,….,n.

In order to link the number expected of malaria cases, E($Y_{1}ǀ\alpha ,\mu_{i}$) = $\mu_{i}$
*i*, with the environmental variables we consider that:(3)$$\mu_{i}=\mu_{i}\left(X_{i}\right)=\exp\left\{\;x^{\prime },\beta \right\}=\exp\left\{\beta_{0}+\beta_{1}x_{I1}+\beta_{2}x_{I2}+\beta_{3}x_{I3}+\beta_{4}x_{I4}+\beta_{5}x_{I5}+\beta_{6}x_{I6}\right\},\;$$
where, $x_{i}=(1,x_{I1}\ldots .,x_{I6)\;}\;$ is a 7-dimensional vector of covariates $\mathrm{and}\;\beta =\left(\beta_{1\;,\;.\;.\;.\;,\;}\beta_{p}\right)$ is a 7-dimensional vector of parameters, for i=1, . . . ,n.

The log-likelihood function for parameters β  is given by:(4)$$l\left(\alpha ,\beta \right)=\overset{n}{\underset{i=1}{\sum }}\log\left(\mu_{i}\right)-\log\left(1+\alpha \mu_{i}\right)+\left(y_{i}+\alpha y_{i}\right)-\log\left(y_{i}!\right)-\frac{\mu_{i}\left(1+\alpha y_{i}\right)}{1+\alpha y_{i}}\;,$$
where $\mu_{i}\;$ given in Equation (2). The maximum likelihood estimates $\hat{\beta }$ of the parameters β  maximize l(α,β),

The maximum likelihood estimates are obtained solving the system of equations given by:(5)$$U(\beta ǀy,x)=\frac{\partial \left(\alpha ,\beta \right)}{\partial \beta }=0\;(3)$$
where U(βǀy,*x*) = $\left(\frac{\partial l\left(\alpha ,\beta \right)}{\partial \beta_{1}},\;.\;.\;.\;,\frac{\partial l\left(\alpha ,\beta \right)}{\partial \beta_{6}}\;\right)$.

However, Equations in (3) do not have solutions. Therefore, we apply numerical methods to solve these equations. Iterative solutions of these equations are the maximum likelihood estimates (MLE) of parameters *β*.

MLE package was used to obtain maximum likelihood estimates in R. Using the stepwise method in the MLE package, we compared different models using the Aitkin Information Criteria (AIC) and reported the results of the model with the smallest AIC as the model with the best fit [15].

## 3. Results

### 3.1. Spatial and Temporal Distribution of Malaria Morbidity

#### 3.1.1. Spatial Distribution of Malaria Morbidity

Malaria prevalence in South Kivu was 7700 cases per 100,000 exposed persons over the past 19 years. Several provinces report between 4980 and 7,00 cases per 100,000 exposed persons. The same trends are observed in the northwestern part (Kinshasa, Equateur, and Nord-Ubangi), the central part (Kasai Central and Oriental), and the southeastern part of the country (Haut-Uélé and South Kivu). The orange color symbolizes the malaria prevalence ranging from 4310 to 4980 cases per 100,000 exposed persons (Figure 1).

#### 3.1.2. Spatial and Temporal Distribution of Malaria Morbidity

The average malaria prevalence for the last 19 years is 13,246 (11,784–14,175) cases per 100,000 people at risk. This prevalence increases significantly at the beginning of the study period (*p* < 0.0001). The year 2002 was the most morbid with 29,138 (12,095–38,305) cases per 100,000 persons exposed to risk. The year 2003 was the second most morbid with 24,466 (12,110–38,873) cases per 100,000 persons exposed (Table 1).

From 2001 to 2002, malaria caused more deaths in families with a case-fatality rate higher than 0.66% (Table 2).

### 3.2. Weather Profile

See also the associated figures for each weather parameter in Appendix A (Figure A1, Figure A2, Figure A3, Figure A4, Figure A5 and Figure A6).

Over the past 19 years, it has rained on average 11.4 ± 2.8 days during the 19 years’ period. High water evaporation was observed throughout the study period, especially in the western part of the country (Kwilu and Kwango), in the center (Sankuru), in the northwest (Mongala) and in the eastern part of the country (North and South Kivu), reaching between 70 and 110 square centimeters of water surface (see map in Appendix A). Relative humidity was also noted to be above 80%. The total quantity of rainfall in mm varies considerably throughout the year. The heaviest rainfall recorded during the study period reached on average a rainfall of over 127 mm. It rains heavily in the northwestern part of the country with total rainfall quantities ranging from 148 to 176 mm (see map in Appendix A).

Describing the temperature variability, it is very useful to summarize them in the attached maps. The average minimum and maximum temperatures are, respectively, 19.6 °C (2.1) and 29.5 °C (2.3) (Table 3).

Windstorms blow heterogeneously across the country (3.8 m/s). Wind speeds of 3.87 to 4.53 m per second at two meters above ground level were recorded in the southern part of the country (Haut-Katanga and Haut-Lomami), in the center of the country (Kasai-Central, Kasai-East, and Sankuru), in the southwest (Equateur and Tshuapa), and in the northeast (Bas-Uélé, Haut-Uélé and Ituri). Wind gusts greater than 5 m/s are observed between August and October (see map in Appendix A).

### 3.3. Climate Predictors Associated with Malaria

This section assesses the main objective of modeling the relationship between malaria morbidity and climate factors between 2001 and 2019 by adjusting for spatio-temporal variation to identify the impact of the climatic factors on malaria morbidity.

#### 3.3.1. Matrix of Correlations

Spearman’s correlation analysis indicates a statistically significant association between evaporation within two meters of the ground and the number of annual malaria cases (*r* = 0.25; *p* < 0.001).

A statistically insignificant association was observed between the rest of the climate variables and the number of annual malaria cases (Figure 2).

#### 3.3.2. Generalized Linear Model of Poisson Regression

All other things being equal, the population in the western region of the DRC on average had 4.92 (exp (1.5936) times as many cases of malaria per 100,000 than the northwestern part of the country’s population. However, the population in the center and the northeast had 2.67 (exp (0.9823) and 2.58 (exp (0.9492) times as many cases of malaria per 100,000, respectively (*p* < 0.0001) (Table 4).

Regardless of other factors, when total evaporation under shelter increases by one unit, the probability of malaria decreases by 2% (*p* < 0.0001). When the average relative soil moisture increased by one percentage point, the risk of malaria morbidity decreased by 0.42% (*p* < 0.0001). Adjusting for other factors, a one-day increase in rainfall resulted in a 7% increase in malaria cases. This relationship was statistically significant (*p* < 0.0001). For each increase in maximum temperature of one degree Celsius, the risk of malaria infestation increased by 1.08 times cases (*p* < 0.0001). When the average wind speed at 2 m above the ground increased by one meter per second, the risk of a malaria episode decreased by 18%. This relationship was statistically significant (*p* < 0.0001) (Table 4).

## 4. Discussion

### Spatial and Temporal Distribution of Malaria Morbidity

Malaria is caused by infection with *protozoan* parasites of the species *Plasmodium. Plasmodium falciparum* is widespread in Africa, while *P. vivax*, *P. ovale*, and *P. malariae* infections are less common and geographically limited (1,2). In DRC, the results observed in this study show an average malaria prevalence of the last 19 years of 13,246 (1,178,383–1,417,483) cases per 100,000 people at risk. This prevalence increases significantly during the whole study period. The year 2002 was the most morbid with 2,913,799 (1,209,451–3,830,456) cases per 100,000 persons at risk.

Most cases (213 million or 93%) were recorded in 2018 in the WHO African region, far ahead of the Southeast Asian region (3.4%) and the Eastern Mediterranean region (2.1%) [3]. Our study showed that Malaria case-fatality in DRC has been decreasing over the past 19 years, from 0.67 (0.43–0.91) deaths per 100 positive test cases in 2001 to 0.11 (0.09–0.14) deaths per 100 positive test cases in 2019. This case fatality varied significantly throughout the study period. From 2001 to 2002, malaria caused more deaths in families with a case-fatality rate higher than 0.66%.

The same observations were made by WHO 2019 showing that on a global scale, malaria incidence declined between 2010 and 2018 from 71 cases per 1000 people at risk of malaria to 57 per 1000. Nevertheless, this decline slowed considerably between 2014 and 2018, with incidence decreasing to 57 per 1000 in 2014 and remaining at a similar level through 2018 [3].

The WHO Africa region alone accounted for 94% of global malaria deaths in 2018. Yet it also accounted for 85% of the 180,000 fewer deaths from the disease compared to 2010 [3]. Nearly 85% of global malaria deaths in 2018 were concentrated in 20 countries in the WHO African region and India. Nigeria alone accounted for nearly 24% of these deaths, followed by the Democratic Republic of Congo (11%), the United Republic of Tanzania (5%), and Angola, Mozambique, and Niger (4% each) (3).

In terms of the correlation between malaria morbidity and mortality and climatic factors, in our study, Spearman’s correlation analysis indicates a statistically significant association between evaporation at two meters from the ground and the number of annual malaria cases (*r* = 0.25; *p* < 0.001). A statistically insignificant association was also observed between the rest of the climate variables and the number of annual malaria cases. In contrast to a statistically significant association between all climate variables studied and monthly malaria cases reported in South Africa where minimum temperature had the highest correlation (*r* = 0.39; *p* < 0.001), followed by total quantity of rainfall at two meters altitude and mean temperature (*r* = 0.35; *p* < 0.001, *r* = 0.35; *p* < 0.001), then relative humidity and maximum temperature with (*r* = 0.29; *p* < 0.001, *r* = 0.25; *p* < 0.001), respectively [16].

This difference could be related to the disparity in transmission levels in Africa, both in intensity and in seasonality and regularity. This disparity has consequences at several levels: biological, immunological, pathological, etc. Subjected to a strong and permanent *plasmodial* infection, the population develops a premunition, a temporary immunity, non-sterilizing and maintained by the infection, itself maintained by the transmission.

It should be noted that South Africa is located in an area where transmission is regular every year, with a long seasonal interruption of about 6 months, linked to the rhythm of the rains. The populations of these areas have a very high degree of resistance to malaria but are not totally free of “moderate” malaria attacks and may be susceptible to severe malaria attacks in case of *plasmodial* infections by foreign strains reflecting the dynamics of host-vector-parasite relationships in different environments according to Wilson’s classifications of epidemiological facies of basic models [17].

There are seasonal variations in the intensity of transmission, but no interruption, however brief. Transmission is generally by *An. gambiae* and *An. funestus*. This mode of transmission is well illustrated by the degraded forest areas of Central Africa (DRC, Congo, Cameroon, etc.).

This approach by Wilson highlights the transmission/malaria disease relationships, taking into consideration the intensity of transmission (chance of infection or frequency of infection), the duration of transmission with its regularity from one year to the next (transmission continues nearly all-year-round or annually recurring season or seasons of malaria transmission), the ecological parameters (altitude, temperature, humidity) with the relative development of immunity. Variations in these different parameters are reflected in variations in the classical malaria indices (*splenic index, plasmodic index*) but above all in malaria morbidity according to age groups and general mortality in the event of an epidemic.

Some authors agree with Wilson’s categorization of epidemiological facies where the DRC/Zaire is located in the Degraded Forest Zone: permanent transmission (=Wilson’s Group I): where malaria is endemic with intense and permanent transmission. There are seasonal variations in the intensity of transmission, but there is no interruption, however brief. Transmission is usually by *An. gambiae* and *An. funestus*. This mode of transmission is well illustrated by the degraded forest areas of central Africa [18].

Our results corroborate several studies. In a study conducted in South Africa, in the local municipality of Mutale, the results show positive correlations between monthly malaria cases and all climatic variables (See Figure A1, Figure A2, Figure A3, Figure A4, Figure A5 and Figure A6 in Appendix A). Mean minimum temperature had the highest correlation coefficient with malaria cases. This indicates that minimum temperature influences malaria transmission more than any other climate variable in Mutale local municipality [17]. On the other hand, the combination of monthly total rainfall and monthly mean minimum temperature, with a lagged effect of two months, were found to be the most significant climate variables in predicting malaria transmission in Mutale municipality with R^2^ = 0.65 compared to R^2^ = 0.54 (mean maximum temperature and rainfall), R^2^ = 0.51 (mean temperature and rainfall), and R^2^ = 0.49 (mean relative humidity and rainfall). This result is similar to that reported by [19], in the study conducted in Mpumalanga province also in South Africa.

The results of our study showed a negative but not statistically significant association between the number of annual malaria cases and geo-climatic factors: wind speed, number of rainy days, maximum and minimum temperature. Our results do not concur with those found by Lindsay et al. showing a negative but statistically significant association between rainfall and malaria cases, which could be due to the fact that the breeding sites were washed away by heavy rains [20]. The significant relationships observed with a lag of more than one month between malaria onset and associated climatic variables, such as rainfall and minimum temperature at two months, suggest that climatic conditions in a given year (early onset of rains, especially those associated with La Niña conditions and tropical cyclones) may affect malaria transmission in the following year. Periods of heavy rainfall combined with low temperatures and high relative humidity promote saturated soil moisture, which prolongs the life of water pockets and may result in the persistence of larval habitats [19,21].

The climatic predictors associated with malaria were identified by Poisson regression. It was found that the population in the western region of the DRC had on average 4.92 times as many cases of malaria per 100,000 than the northwestern part of the country’s population. However, the population of the center and the northeast have respectively 2.67 and 2.58 times as many cases of malaria (*p* < 0.0001). Malaria transmission is influenced by many factors, including malaria vector abundance, mosquito survival rate and longevity, parasite development rate in mosquitoes, mosquito biting rate, and human susceptibility to parasites governed by human behavior and immunity [16].

We believe that further studies including other aspects may shed more light on this point. Our study specifically modeled malaria morbidity as a function of geo-climatic factors. Our study also showed that when total evaporation under shelter increases by one unit, the probability of malaria decreases by 2% (*p* < 0.0001) and malaria increases by 97% and 94%, respectively, in the southeast and west regions compared to the other regions (*p* < 0.05). When the average relative soil humidity increased by one percent, the risk of malaria morbidity decreased by 0.42% (*p* < 0.0001).

Adjusting for other factors, a one-day increase in rainfall resulted in a 7% increase in malaria. This relationship is statistically significant (*p* < 0.0001). Our results are similar to those found by [16]. According to the authors, mean monthly rainfall for their study showed a significant positive correlation with malaria cases with a two-month lag in all zones, while mean maximum temperature showed a significant positive correlation with malaria cases in two zones, the highlands and riverine areas.

Previous studies examining the relationship between climate and malaria have found lagged associations between climate variables (temperature and rainfall) and malaria cases over periods ranging from weeks to months [10,11]. In our study, we also forecast that with each increase in maximum temperature by one degree Celsius, the risk of malaria infestation is increased by 1.08 (*p* < 0.0001).

According to a 2007 study, periods of unusually high rainfall, altered humidity, or warmer temperatures can lead to changes in malaria distribution and duration, as well as increased transmission, even in areas with strong control. Consistent with current findings, the authors documented rainfall and temperature as the major factors in the variation of malaria cases in Africa, while acknowledging the complexity of some climatic factors. The difference in the environmental relationship with malaria cases in different areas is attributed to variations in environmental factors between areas [22]. When the average wind speed at 2 m above the ground increased by one meter per second, the risk of a malaria episode decreased by 18%. This relationship was statistically significant (*p* < 0.0001). Temperature affects the development of malaria; the parasite does not develop below 18 °C and above 40 °C [23,24].

An increase in temperature can reduce the time of production of new generations and shorten the incubation period of the parasite in mosquitoes. Sporogonic cycles last approximately 9–10 days at 28 °C, but temperatures above 30 °C and below 16 °C negatively impact parasite development [25]. The highest proportion of vectors surviving the incubation period is observed at temperatures between 28 and 32 °C [26].

In this study, the average maximum temperature recorded was 26.8 °C, ranging from 22.3 to 31 °C, suggesting that DRC is the ideal country for malaria breeding. In the present study, the minimum temperature was below 18 °C from week 10 to week 40, coinciding with a marked reduction in malaria occurrence. In this study, the mean temperature was found to be a significant predictor of malaria occurrence, similar to studies conducted in South Africa (19) et and Burundi [27]. Relative humidity (RH) also plays a role in malaria episodes, and mosquitoes become more active as humidity increases. If the average monthly RH is less than 60%, the mosquito’s life is thought to be so short that very little or no malaria transmission is possible (28,29). In this study, relative humidity was 72.1% and only four weeks of the year had relative humidity below 60%, implying that humidity does not limit the presence of malaria in DRC. Similar results were also reported in a study in Ghana [28,29] (See also Figure A1, Figure A2, Figure A3, Figure A4, Figure A5 and Figure A6 in Appendix A).

Wind speed was found to be a significant influence in the occurrence of malaria in Nigeria [29,30]. In this study, wind speed was not found to be a significant predictor of malaria occurrence in Chimoio. Visibility was not found to be a significant predictor of malaria occurrence as shown in studies conducted in Nigeria [29] and South Africa [19]. Most anopheles mosquitoes are crepuscular (active at dusk or dawn) or nocturnal (active at night) [19].

The frequency of foggy days was found to have a positive effect on malaria incidence the following year [31]. The World Health Organization indicates that the increase in temperature and precipitation linked to climate change will increase the number of mosquitoes present in the coldest areas where people, not being used to these diseases, have very little resistance to them. This is in line with a study published by the Lowy Institute in Sydney, Australia, indicating that the prevalence of malaria could be 1.8 to 4.8 times greater in 2050 than in 1990, according to forecasts.

## 5. Conclusions

Climatic factors are an important determinant of a variety of vector-borne diseases, many enteric diseases, and some water-borne diseases. The relationship between year-to-year variations in climate and infectious diseases is particularly evident in vulnerable populations and where climatic variations are marked. Malaria occurs only in tropical and subtropical regions.

The geo-climatic predictors most associated with malaria identified by the Poisson regression were geographic location (western, central and northeastern region of the country), total evaporation under shelter, maximum daily temperature at two meters altitude. Finally, the average number of malaria cases increased positively as a function of the average number of rainy days, the total quantity of rainfall and the average daily temperature. Based on our findings, it is important to set up a warning system integrating the monitoring of rainfall and temperature trends and the early detection of anomalies in weather patterns in order to forecast potential large malaria morbidity events.

## Figures and Tables

**Figure 1 ijerph-19-03811-f001:**
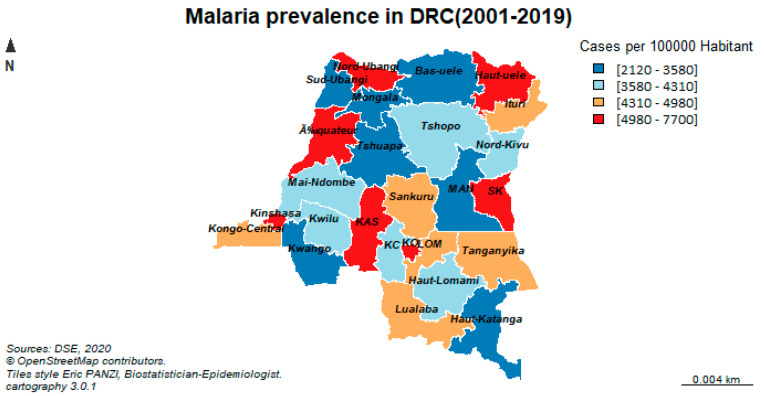
Geographic distribution of malaria prevalence in the DRC from 2001 to 2019.

**Figure 2 ijerph-19-03811-f002:**
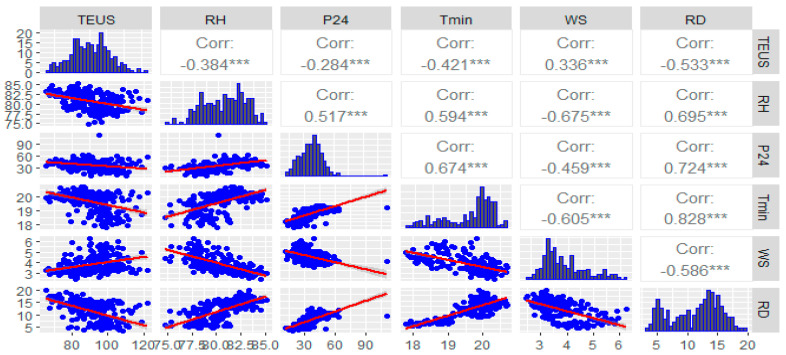
Scatterplot matrix showing the correlation study variables. The values of each variable were normalized by the transformation to its empirical percentile. See the legend of the study variables in the Appendix A. *** Statistically significant *p*-value of <0.0001.

**Table 1 ijerph-19-03811-t001:** Malaria prevalence rate (per 100.000 persons at risk) in DRC, 2001–2019.

Year	n			Prevalence	95% Confidence Interval	
		Total Cases	Person-Years at Risk		Lower Bound	Upper Bound
**2001**	**26**	29,310	575,363	509	2918	7182
2002	26	173,182	594,350	29,138	12,095	38,305
2003	26	150,213	613,964	24,466	12,110	38,873
2004	26	42,518	634,224	6704	4407	8317
2005	26	80,413	655,154	12,274	7332	14,727
2006	26	55,812	676,774	8247	5065	10,857
2007	26	56,627	699,108	8100	5248	9887
2008	26	56,321	722,178	7799	5543	9439
2009	26	73,720	746,010	9882	7074	11,717
2010	26	85,153	770,628	11,050	7854	12,937
2011	26	89,301	796,059	11,218	8249	13,793
2012	26	93,771	822,329	11,403	9165	13,859
2013	26	98,428	849,466	11,587	9420	13,569
2014	26	109,644	877,498	12,495	10,555	15,395
2015	26	118,909	906,456	13,118	10,669	16,136
2016	26	141,464	936,369	15,108	13,126	18,197
2017	26	159,141	967,269	16,453	13,168	22,395
2018	26	167,609	998,169	16,792	14,362	20,382
2019	26	188,362	1,030,056	18,287	15,794	23,097
Total	494	1,969,897	14,871,424	13,246	11,784	14,175

The *p*-value was obtained from the chi-square test. Note: Figures in bold indicate malaria prevalence (*p*. 100,000 persons at risk) significantly higher than the mean for a period (*p* < 0.0001).

**Table 2 ijerph-19-03811-t002:** Malaria case-fatality rate (Deaths per 100 cases) in DRC, 2001–2019.

Year	n	Lethality	95% Confidence Interval	
			Lower Bound	Lower Bound
**2001**	**26**	0.67	0.43	0.91
2002	26	0.68	0.53	0.83
2003	26	0.52	0.38	0.67
2004	26	0.45	0.34	0.56
2005	26	0.33	0.25	0.42
2006	26	0.33	0.26	0.40
2007	26	0.33	0.26	0.39
2008	26	0.32	0.25	0.40
2009	26	0.28	0.22	0.34
2010	26	0.25	0.17	0.32
2011	26	0.29	0.21	0.37
2012	26	0.27	0.19	0.36
2013	26	0.29	0.19	0.40
2014	26	0.27	0.20	0.34
2015	26	0.23	0.18	0.29
2016	26	0.20	0.16	0.24
2017	26	0.15	0.11	0.18
2018	26	0.12	0.09	0.15
2019	26	0.11	0.09	0.14
Total	494	0.32	0.30	0.35

The *p*-value was obtained from the chi-square test. Note: Bolded numbers indicate malaria case-fatality rates that are significantly higher than the average for a given period (*p* < 0.0001). Malaria case-fatality in DRC has been decreasing over the past 19 years, from 0.67 (0.43–0.91) deaths per 100 positive test cases in 2001 to 0.11 (0.09–0.14) deaths per 100 positive test cases in 2019. This case fatality varied significantly throughout the study period (*p* < 0.0001).

**Table 3 ijerph-19-03811-t003:** Weather parameters in DRC, 2001–2019 (26 provinces xX 19 ans), n = 494.

Weather Parameters	Mean	sd
Rainy day (day/month), mean(sd)	11.4	2.8
Total evaporation under shelter (pitcher)	90.5	20.5
Humidity (%)	80.7	5.6
Total rainfall (mm)	127.6	35.5
Precipitation of 24 h	37.4	12.3
Max. temperature (°C)	29.5	2.3
Min. temperature (°C)	19.6	2.1
Wind speed at 2 m from the soil (m/s)	3.8	0.5

**Table 4 ijerph-19-03811-t004:** Poisson Regression β coefficients by selected study characteristics.

Variables/Distance	β	*p*-Value
(Intercept)	127,424	0.0001
ln (Time)	10,000	
Regions		
Northwest (reference class)	10,000	
West	15,936	0.0001
Central	0.9823	0.0001
North-East	0.9492	0.0001
Evaporation	−0.0191	0.0001
Soil moisture	−0.0043	0.0001
Average number of rainy days	0.0645	0.0001
Total rainfall quantity	0.0002	0.0001
Precipitation 24 h	−0.0010	0.0001
Minimum temperature	−0.1302	0.0001
Maximum temperature	0.0726	0.0001
Wind speed	−0.2017	0.0001
sigma	10,800	0.0001

## Data Availability

We exploited the databases of the Directorate of Epidemiological Surveillance and the Mettelsat database for meteorological data centralizing data from 26 provincial health directorates of the DRC from 2001 to 2019. Climatic data including total rainfall quantity, temperature, wind speed, total evaporation under shelter (piche), average relative humidity, were measured permanently by means of different meteorological stations in the DRC.

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
