# Peer review of "Geo-Climatic Factors of Malaria Morbidity in the Democratic Republic of Congo from 2001 to 2019"

_ijerph, 2022, doi:10.3390/ijerph19073811_

Round 1
Reviewer 1 Report
Nothing to suggest after the review made by the authors.
Reviewer 2 Report
This study provides novel insights about the impacts of climatic variables on malaria prevalence. This is a great effort of aggregating data on malaria prevalence and climatic variables to understand their effect on malaria morbidity. The document needs a lot of work before acceptance for publication in the International Journal of Environmental Research and Public Health. Please see my detailed comments and my main concerns:
- Title: I don't know if it is a good idea to specify the model in the title. First, the authors did not show the distribution of the data either in the main document or in the appendix (e.g., histogram). Is the Poisson model the best model for this kind of data? I did not observe any kind of model comparison in the document. What about a negative binomial model, or a ZIP model?
- Abstract:
- Line 30: "a one day significant increase of rainfall" (What does mean a significant increase of rainfall??)
- Line 31: What exactly means "there was overwhelming evidence of associations between ..........." Provide some values. Overwhelming is too vague.
- Improve the conclusions in the abstract. Right now it is a repetition of the results.
- Keywords: Avoid repeating the same words from the title.
3. Introduction:
- First of all, be sure to use italics for scientific names; check through the manuscript.
- The focus of the first paragraph should be to introduce the study. For example, how environmental variables affect malaria morbidity, malaria prevalence, or malaria in general.
- Next, organize the following paragraphs keeping in mind that your goal is to keep the reader flowing with the story. For example, the second paragraph could be your lines 57-65. The third paragraph introduces general information about malaria. Next, talk about the kind of data you have aggregated and possible best models to analyze it. Finally, summarise how the manuscript is going to be organized (move here Lines 67 -74 and improve them).
4. Methods:
- You have a good amount of data that could be better used it. For example, you could generate new covariates from this data-set; Time lags of 3, 6, 9, and/or 12 weeks prior to the sample days to see the effect of temperature, rainfall, etc. previous to the study day.
- How exactly the data was aggregated into 494 observations. Please provide as much detail as possible on how you aggregated and why did you aggregate these data (If there is a pages limitation, include a section in the appendix).
- Organize the statistical methods in a clearer way. Add references for the use of the mean-value imputation technique.
- What exactly is the spatio-temporal analysis? describe the methods, add references. very poor writing.
- About the modeling of the impact of climate factors on malaria morbidity, please improve the description of the model, review some other papers on the topic and see how do they explain the models. Furthermore, I don't see any kind of model selection. For example AIC values. How do you find the best model for your data? How are you dealing with correlated covariates? In your matrix of correlations, some of your covariates are highly correlated; this could impact the estimates of the coefficients and produce greater standard errors (sometimes you can drop one of the correlated variables from the model). So, please check for collinearity. A lot of work is needed in the methods section.
5. Results:
- How useful is table 3? It would be better to present linear plots showing the trends of rainfall, temperature, evaporation from 2001-2019.
- Fix the labels in the matrix of correlations. They are cut in many places.
- Line 288: You said the heaviest rainfall recorded during the study period is 127 mm but in table 3 is presented as the mean. Please double-check.
- Table 4: please review how to present the results from a glm model. Columns 2 and 3 are the same information (one in the log scale)
6. Discussion y Conclusion
- Avoid repeating the same information in the results and the discussion (e.g., lines 337 - 341).
- Once the methods and results sections are improved please come re-write your discussion and conclusion sections.
Minor comments:
Line 167: missing r in recode
Line 385: missing l in mode
Table 4: very hard to read the text in column 1: use a different color for the background and the font.
Appendix: Y labels in all the graphs are using numbers and months but the last three months use x,y,z. Also, very hard to read the names on the maps.
If possible, please send me your code and a sample of the data.
Reviewer 3 Report
Please, find attached my comments.

Round 2
Reviewer 2 Report
There is some improvement in the writing and presentation of the document. However, the methods and data analysis were not addressed in the review of the document. For example, in line 158 the authors mentioned that the mean and variance are almost the same and because of this they could use a Poisson regression model, but they used instead the Famoye's restricted generalized Poisson distribution which is used for overdispersed data.
Second, how do they know this is the best model without a model evaluation. They mentioned they used AIC values to evaluate the models but I don't see what models do they evaluated? They are only showing one model. Please, compare different models. For example compare the Poisson regression, the negative binomial (Poisson-gamma model), the famoye's generalized Poisson model, etc using AIC or BIC values. Another alternative is to validate your model to see if this is the best model. Use part of the data as a training dataset and the rest of the data as a testing dataset. It is unacceptable to say this is the best model because it was used in similar studies, you need to prove this is the best model for your data set.
Furthermore, please review correlated variables, your correlation matrix shows correlated variables (see your values of 0.707, 0.918, -0.673, etc.). Also, labels are still cut-off in the legend of the correlation matrix.
Before acceptance, this study needs a lot of work in the methods and data analysis.
For further advice, please feel free to email me your dataset.
Please review the paper Regression Models for Count Data in R by Zeileis et al. It provides nice examples of models for count data and model comparison. Also, check the book of Alain F. Zuur entitled Mixed Effects models and extensions in ecology with R.
Reviewer 3 Report
Please, find attached my comments on the paper "Climatic factors of malaria morbidity in the Democratic Republic of Congo from 2001 to 2019: Generalized Poisson regression model".

Author Response
see attached

This manuscript is a resubmission of an earlier submission. The following is a list of the peer review reports and author responses from that submission.
Round 1
Reviewer 1 Report
The paper presents a generalised linear model for forecasting malaria epidemics taking into account climatic and spatio-temporal variations. In addition, this paper contains a model to assess the impact of climatic factors on malaria morbidity.
The framework provided in this paper can improve the understanding of geo-environmental factors' impact and offers public health authorities insight to set alarm systems. The paper is written in understandable English, but some details are not well explained, and several inconsistencies are easy to spot. Besides, an English language review is needed.
Overall, the paper is fascinating and useful to the modelling community and public health in general.
Specific comments:
- In line 59, where we can read, "In the past, several authors have already highlighted the existing relationship between climatic variations and malaria endemicity(3)", reference (3) does not match the sentence. I suspect it should be reference (4). Anyway, authors should provide a broad list of references on this topic. There are plenty of examples available in the most recent years.
- In line 65 authors state: "Establishing a relationship between the environment and health still remains a challenge for scientific knowledge because the causal chain between environmental factors and the resulting disease is complex (6)." More elaboration is needed to clarify the meaning of "complex".
- Line 74: a new paragraph is not necessary.
- Line 111: DPS acronym was not defined beforehand.
- In section 2.2 authors explain that data comprises 26 provinces, 19 years, and 52 weeks. Then they state the data was aggregated to 494 observations but do not explain how. I suspect the authors collapsed the 52 weeks into yearly totals. Is that so?
- In line 155 authors state: "The imputation technique is used to fill in missing data." I wonder which imputation technique was used, as there are many!
- In section 2.4.3. the model is not clear and accurately enunciated. Closed-form of function f is not defined. The regression coefficients vector is absent from equation (2). Two types of predictors are implicit in equation (2), but one group of six climate predictors is mentioned in the text.
- In Figure 3.1.1. legend suggests the prevalence is measure in cases per million inhabitants. However, in the text, it is referenced as cases per 100 000 people.
- In line 190, the sentence " In the northwestern part, there is Kinshasa, Equateur and North-Ubangi" has no consequence.
- Table 3.1.1. can be replaced by a plot to exhibit the temporal trend easily.
- Idem for Table 3.1.2.
- Table 3.2 doesn't have column names. Analysis of table values is done by a value +/- a second value (it seems to come from the table's second column). What is that? Standard deviation? Confidence interval semi-range?
- At the end of section 3.3.1 authors remark on the difference between multi-linearity and colinearity. This note makes 3.3.1 useless!
- There are several places where English is mixed with French!
- In line 297: " moisture increased by one percent" should be replaced by "moisture increased by one percentage point".
- Figures referred to in paragraphs starting in lines 291 and 295 should be revised.
- Figures about malaria mortality referred to in the Discussion are not coherent with the table.
- In several places, authors refer to "generalized linear regression of fish", which is, in fact, a Poisson regression.
- References in the text are not consistently done in the same way.
- More elaboration is needed on the alarm system referred to in the abstract and introduction.
Reviewer 2 Report
Please see my comments attached.

Reviewer 3 Report
This manuscript attempts to identify environmental factors associated with increased malaria morbidity in the
Democratic Republic of Congo. In itself, this is a valid goal. Unfortunately, the composition due to language difficulties render this paper suspect. There are just too many language mistakes to allow a trustworthy transmission of the information. Two mistakes demonstrate this problem. The first is the source of the data, which is repeatedly referred to as the Environmental Surveillance "direction". It is hard to find this source, but I suspect it is a directive, not a direction. The second is more serious. The authors refer repeatedly to a "linear fish regression" or "general linear regression of fish". This is actually a reference to Poisson regression. "Poisson" is the name of the man who developed the test. It does indeed mean "fish" in French, but it is the man's name. It may be that the authors ran this through GoogleTranslate or a similar software package, but this is not an accurate practice. In order to make this paper acceptable, it must be completely rewritten by someone who has a deep understanding of both the subject and English composition. In its current state, the paper does not present the subject in a way that can be clearly understood.